# Study on the SPCC and CFRTP Hybrid Joint Performance Produced with Additional Nylon-6 Interlayer by Ultrasonic Plastic Welding

**DOI:** 10.3390/polym14235235

**Published:** 2022-12-01

**Authors:** Tai Wang, Kiyokazu Yasuda, Hiroshi Nishikawa

**Affiliations:** 1Materials and Manufacturing Science Division, Graduate School of Engineering, Osaka University, Osaka 565-0871, Japan; 2Joining and Welding Research Institute, Osaka University, Osaka 565-0871, Japan

**Keywords:** SPCC-CFRTP hybrid joint, intermediate layer, ultrasonic plastic welding, preheating, joining mechanism

## Abstract

Due to the high degree of dissimilarity in physicochemical properties between metal and carbon fiber, it presents a tremendous challenge to join them directly. In this paper, cold rolled steel (SPCC) and carbon fiber reinforced thermoplastic (CFRTP) chopped sheet hybrid joints were produced with the addition of Nylon 6 (PA6) thermoplastic film as an intermediate layer by the ultrasonic plastic welding method. The effect of ultrasonic welding energy and preheating temperature on the hybrid joint microstructure and mechanical behavior was well investigated. The suitable joining parameters could obtain a strong joint by adding the PA6 film as an intermediate layer between the SPCC and bare carbon fibers. Microstructural analysis revealed that the interface joining condition between the PA6 film and the SPCC component is the primary reason for the joint strength. The crevices generated at the interface were eliminated when the preheating temperature arrived at 200 °C, and the joint strength thus significantly increased. The lap shear test results under quasi-static loading showed that the welding energy and preheating temperature synergistically affect the joint performances. At 240 °C, the joint strength value reached the maximum. Through the analysis of the microstructure morphology, mechanical performance, and the failure mechanism of the joint, the optimized joining process window for ultrasonic plastic welding of SPCC-CFRTP by adding an intermediate layer, was obtained.

## 1. Introduction

To meet the requirement of the sustainable society development strategy, and a carbon-neutral target in the world, automobile manufacturing factories have a great deal of interest in finding alternative solutions to produce lighter vehicles through material substitution [1]. Carbon fiber reinforced thermoplastic (CFRTP) has attracted much attention as the next-generation of structural material. Carbon fibers have better wear resistance, toughness, and tensile strength than glass fibers. In addition, CFRTP presents high productivity compared with epoxy-based carbon fiber-reinforced thermosetting plastics, due to quick injection molding and press molding techniques [2,3,4]. The metal-CFRTP hybrid structure is a promising method that can effectively increase vehicles’ lightweight potential and meet safety performance and mechanical properties [5,6]. However, due to the high degree of dissimilarity in physicochemical properties between metallic materials and CFRTP, the possibility of the joining process is indeed a great challenge to achieve the intended results [7,8]. 

Traditional joining technologies for metals and CFRTP include mechanical joining, adhesive joining, direct welding, or combining these methods [9,10,11]. These joining technologies have benefits, drawbacks, and their own application occasions [10,12,13]. Conventional mechanical joining employs bolts or rivets associated with a long joining time due to hole drilling and fastening operations, which can cause damage to carbon fibers and stress concentration at the joining zone [9,14]. Adhesive joining has a high requirement of surface pre-treatment, leading to a long curing time and expensive causes [15]. Other disadvantages of adhesive joining are the low durability of joints and harmful environmental emissions [16]. Generally, these joining technologies can produce robust processes and good joint properties [17,18]. However, the common disadvantages, such as relatively long welding cycles, expensive equipment, and intricate automation, have stimulated researchers to look for further simplified investigations [19,20,21]. 

Meanwhile, the above methods can only be applied to joining the CFRTP sheet with cured resin on the surface. For some particular purposes of CFRTP sheets, carbon fibers are exposed on the surface, which makes it impossible to produce the metal-CFRTP hybrid joint through a direct joining method, due to the vast physical and chemical differences between carbon fibers and metallic materials. To solve this problem, some researchers demonstrated the feasibility of the joining method by adding thermoplastic film co-cured as an intermediate layer with the composite parts [7,22]. The existence of the thermoplastic film can essentially reduce the performance difference between metal and carbon fiber and generate effective joining. It also proved successful in welding two thermosetting matrix composite adherents, as demonstrated by Lionetto et al. [23]. This solution is based on the benefits of weld-bonding technology to improve the mechanical performance of the hybrid joints [24]. During the joining process, the intermediate layer is melted or softened under the action of external energy, and the subsequent solidification of the layer will supply adhesive force between the composite parts, which contributes to the good mechanical performance of the joint. 

K. Nakata et al. [25] researched the joining mechanisms for friction stir welding of CFRTP and metallic materials by adding an additional intermediate layer. He pointed out that the functional group force was generated at the interface between the metal and thermoplastic. Mitschang et al. [7] investigated induction welding of metal-composite hybrid joints with an additional polymeric film interlayer and found that adding the film improved the shear strength by approximately 15% for AlMg3-CF-PA66. Esteves et al. [22] investigated the joining feasibility of Al-CFRTP by the friction spot joining method with an additional interlayer and optimized the joining process. These results show the feasibility of metal-to-carbon fiber joining by adding an intermediate layer. Furthermore, using a thermoplastic film exempted the required curing time of an adhesive compared to the currently used industrial adhesives.

To achieve the intended purpose with an economical joining method, ultrasonic plastic welding (UPW) technology with a heater system was applied to investigate the weldability and welding process of the metal-CFRTP hybrid joint in this research. In addition, PA6 film was selected as the thermoplastic intermediate layer in this research due to the low crystalline characteristic and good compatible ability with carbon fibers and metals. In the UPW process, the local temperature increases quickly, leading to softening or melting of the PA6 layer, which is displaced from the welding zone underneath the pressure applied by the horn. Meanwhile, melted PA6 may fulfill the wetting on the metal surface, and the adhesion contact between metal and CFRTP is realized. Therefore, to illustrate the influential factor on the hybrid joint performance and gain the feasible welding process, ultrasonic welding energy and preheating temperature were investigated in this research, which is the primary heat source generated to melt the PA6 film. The role of welding energy and preheating temperature, as well as the synergetic effect, was analyzed according to the joint mechanical performance, microstructure morphology, and failure mechanisms of the joint.

## 2. Materials and Experimental Methods

### 2.1. Materials 

#### 2.1.1. Cold Rolled Steel (SPCC)

SPCC is a widely used metallic material for structural parts and inner body panels in automotive manufacturing. Therefore, the 0.8 mm thickness SPCC plates were employed as the metal part of the hybrid joint in this research. Before joining, the SPCC sheets were machined with the dimension of 15 mm × 70 mm and then cleaned in an ultrasonic cleaner with acetone solution for 5 min to remove the organic containments and dust on the surface. In this study, the SPCC sheets were joined with the received condition without any further surface pre-treatment. A laser scanning microscope (VK-9700, Keyence, Japan) was applied to gain the numerical value of its surface roughness parameter (*Ra*), and the arithmetic average is around 21 μm, as shown in Figure 1.

#### 2.1.2. CFRTP Chopped Sheet

Carbon fiber reinforced thermoplastics (CFRTP) chopped sheet with a nominal thickness of 0.5 mm, supplied by Fukubi manufacturer Japan, was applied as the polymer part in this research. The typical properties are listed in Table 1. This CFRTP chopped sheet is manufactured by randomly distributing the thin layer of prepreg cut into strips on intermediate matrix resin, as depicted in Figure 2a. The prepreg is a sheet-like intermediate material in which PA6 thermoplastic resin is impregnated into a carbon fiber system. Figure 2b shows the cross-section of the CFRTP chopped sheet. The distribution of carbon fibers in the chopped sheet is locally regular according to the chopped prepreg, and the diameter of the carbon fiber is around 7 μm shown in Figure 2c. Due to carbon fibers oriented in all directions, the chopped sheet has pseudo-isotropic properties and can be laminated without worrying about the direction of carbon fibers. In addition, carbon fibers are exposed on the sheet surface, which causes carbon fibers easy to move and leads to good formability of the CFRTP chopped sheet. Therefore, this CFRTP chopped sheet is excellent for three-dimensional shape manufacturing. 

#### 2.1.3. Nylon-6 (PA6) Film 

PA6 is a customarily used engineering plastic with excellent mechanical properties and resistance to chemicals and solvents [26,27]. Table 2 lists the common properties of the PA6. According to Natália M et al. research results [23], the film thickness directly influences the formation of the macroscopic mechanical anchoring between the components, and 100 μm is the most appropriate value to obtain hybrid joints characterized by a high single lap shear strength. Thus, this research selected 100 μm thickness PA6 film as the intermediate layer between the SPCC and CFRTP chopped sheet. 

### 2.2. Experimental Methods

#### 2.2.1. CFRTP-SPCC Joint

The SPCC-CFRTP hybrid joint designed in this research contains three components: the top layer CFRTP chopped sheet, the middle layer PA6 film, and the bottom layer SPCC metal substrate, as schematically shown in Figure 3a. Figure 3b depicts the hybrid joint’s cross-section structure and each component’s thickness before joining. In the dissimilar materials joining process, the interface situation is vital in characterizing the joining condition and the joint quality. Thus, two interfaces were defined to analyze the joining state and mechanisms: interface 1 between the CFRTP chopped sheet and the PA6 film and interface 2 between the PA6 and the SPCC metal substrate, as illustrated in Figure 3c. The interspace between the two interfaces existed apparently before the joining, indicating no interaction between the joining components.

#### 2.2.2. Ultrasonic Plastic Welding System

Ultrasonic plastic welding (UPW) is a standard technology used for thermoplastic material joining. The mechanical oscillation of the UPW works perpendicular to the workpieces, which is beneficial for the flowing and spreading of the molten thermoplastics [28]. UPW has several significant advantages compared to other joining methods, such as a fast, flexible, and effective joining process, minimum top surface damage compared to friction stir joining, high structural reliability, minimum contamination risk of charred particles, and friendly to the environment [29]. Significantly, it can reduce costs and is more feasible for the industry’s application [30]. In addition, a high-power digital hot plate (AS ONE Corporation, HP-1SA) as the preheating platform was applied with the UPW system to investigate the influence of the preheating temperature on joint performance. The heating temperature was set at the predetermined value before joining, and an infrared spot sensor (KEYENCE) confirmed the sample surface temperature, so the preheating temperature could reach the pointed value. The joining was conducted 10 s later after the temperature kept stabilized. The whole system is graphically shown in Figure 4, and the right table lists the standard joining parameters. 

In the UPW process, energy weld mode was selected to produce the SPCC-CFRTP hybrid joints. The four controllable joining parameters in energy weld mode are welding force, vibration amplitude, joining time, and energy, which influences the joining mechanisms and, consequently, the joints’ microstructure and performance. Welding force ensures the intimate contact behavior between the joining components and controls the deformation of the PA6 film and CFRTP chopped sheet. In contrast, higher welding force may lead to a large amount of flowing of the molten PA6 film and severe deformation of the CFRTP chopped sheet. Thus, the joining pressure was kept at 100 N for all experiments. Welding energy is the main factor that can affect the hybrid joint properties. Therefore, the influential role of the welding energy on the joint was investigated from 300 J to 1500 J. The welding time is related to the welding energy, and their relationship is shown in Figure 5. For other parameters, such as the vibration amplitude, trigger force, and hold time, they were fixed in the whole welding process, and the values were shown in Figure 4. 

#### 2.2.3. Design of the Experiments

In this research, welding energy and preheating temperature are determining factors for heat generation in the joining process, which affects the condition of PA6 film and is responsible for joint performance. Therefore, this research mainly investigated the effect of welding energy and preheating temperature on joint performance. The welding energy ranged from 300 J to 1500 J, and the selected preheating temperature values are 20 °C (room temperature), 50 °C, 80 °C, 120 °C, 160 °C, 200 °C (melting beginning point of PA6 film), and 240 °C (full melted temperature of PA6 film), respectively. 

### 2.3. Characterization

#### 2.3.1. Joint Microstructure Analysis 

The joint interface joining condition obtained under different welding parameters was analyzed by scanning electron microscope (SEM). The joints were previously cut from the middle by a little cutting machine, then embedded in cold-curing epoxy resin, followed by standard grinding and polishing procedures. Finally, the samples were sputter coated with a thin layer of gold for 1 min before the SEM observation. 

#### 2.3.2. Mechanical Testing

The lap shear test evaluated the mechanical performance of the joint. The tests were carried out in a Zwick/Roell 1478 universal tensile tester based on standard ASTM D3163-01, with a 6 mm/min speed at room temperature. The dimension of the hybrid joint is shown in Figure 6. Three replicates for each joining condition were tested, and the average of the three measurement results was reported as the joint strength. 

## 3. Results and Discussion 

### 3.1. Effect of UPW Energy at Room Temperature 

UPW energy is the primary heat source for softening or melting the PA6 film at room temperature. The joint microstructure obtained under different welding energy was observed utilizing the SEM. Figure 7 shows the joint’s microstructure produced under 500 J welding energy at room temperature. In Figure 7a, the CFRTP chopped sheet, PA6 film, and SPCC metal parts are easy to identify due to joint components being separated and keeping the original status. Consequently, the boundary of interfaces 1 and 2 can be observed clearly in Figure 7b, which indicates the ineffective joining between the joining components. In addition, some crevices were found at interface 2, further verifying the insufficient heat generation generated by 500 J welding energy at room temperature. With the same condition, when the welding energy was 300 J, it could not produce the hybrid joint either.

When the welding energy reached 700 J, the fusion phenomenon partly occurred at interface 1, as shown in Figure 8b. However, the interface 1 boundary still existed for most welding areas, and there was an interspace between the PA6 film and SPCC metal, filled by the cold-curing epoxy resin. Figure 8a shows the edge morphology of the joint. Joining components can be identified, and they did not cause any deformation of the CFRTP chopped sheet and PA6 film. This joining result led to insufficient joint strength, only 30 N around when tested under the lap shear test.

When the welding energy reached 900 J, the carbon fiber bundles were squeezed to the joint edge accompanied by the PA6 film melting, as shown in Figure 9a, which is different edge morphology from Figure 7a and Figure 8a. Meanwhile, the PA6 film fused with the CFRTP sheet, and interface 1 disappeared, as shown in Figure 9b. It suggests that the high oscillation under 900 J welding energy can cause the melting and interdiffusion at the boundary of the PA6 film and the CFRTP chopped sheet. Efficient cohesion is guaranteed between the PA6 film and CFRTP chopped sheet. This phenomenon can be explained by the fact that during the UPW process, PA6 film melted rapidly under the action of welding energy. Then, the bare carbon fibers on the CFRTP chopped sheet surface wrapped by the melted PA6 film were squeezed from the center to the joint edge under the action of the welding force applied by the horn. However, for interface 2, some crevices still occurred, indicating that the molten PA6 film did not fully fill the SPCC surface. It also needs to mention that the carbon fibers can remain at the initial status after the welding process under 900 J welding energy. 

The disappearance of interface 1 under 900 J welding energy proves that effective joining between the PA6 film and CFRTP can be achieved by improving the welding energy. However, the crevices at interface 2 were not fully resolved yet, as shown in Figure 10. Thus, the welding energy continues increased attempt to solve this problem. When the welding energy reached 1100 J, the number of carbon fibers and molten PA6 film squeezed to the joint edge increased in Figure 10a. Interface 1 disappeared in the same way as Figure 9. In addition, carbon fibers were observed to have been damaged due to the strong ultrasonic oscillation under 1100 J welding energy. Cracks appeared at the cross-section of the carbon fiber perpendicular to the ultrasonic vibration direction, which is observable in Figure 10c. Meanwhile, the crevices still existed at interface 2.

To further verify the high welding energy on the microstructure of the joint interface at room temperature, the welding energy was increased to 1300 J and 1500 J. Figure 11 and Figure 12 show the related microstructures, respectively. When the welding energy reached 1300 J, the squeezing phenomenon and the damage to the carbon fibers got more serious, as shown in Figure 11a,c. When the welding energy reached 1500 J, more serious squeezing and damage phenomena happened in Figure 12a,c. However, the crevices at interface 2 still cannot be eliminated even though under the high welding energy that can damage the carbon fibers and squeeze almost all melted PA6 film and carbon fibers out to the joint edge. 

The crevices’ occurrence was considered a joining defect that led to poor joint properties. And from the above experiments, it was found that this joining defect cannot be eliminated by solely increasing welding energy at room temperature. Microstructural analysis at interface 2 illustrates that the crevices always occurred at the depression area of the SPCC surface, as schematically described in Figure 13. During the welding process, the bulge area of the SPCC surface was more feasible to bond with the PA6 layer due to the direct connection. In contrast, the depression areas of the SPCC surface cannot be filled in a dramatically welding process. The main reason is that UPW technology is a high-speed joining technology. Therefore, PA6 film dramatically reached a high-temperature degree and then solidified in a short period. Thus, the short welding time and high cooling rate cannot enable a sufficient spreading of the molten PA6 film at the depression area of the metal surface, which leads to the generation of crevices at interface 2.

In addition, to investigate the effect of welding energy on joint properties at room temperature, the thickness variation of the PA6 film and CFRTP chopped sheet during the welding process was also measured under different welding energy, shown in Figure 14, which can be seen as a characterization to evaluate the effect of the welding energy at room temperature. It appears that the PA6 film thickness was kept at 100 μm before 500 J welding energy, proving that the generated oscillation heat was insufficient to melt the PA6 film and produce the hybrid joint. However, for the CFRTP sheet, the thickness showed a gradual decrease trend with the energy increasing at the beginning. This is mainly due to the interspace among the CFRTP chopped sheet. Under the ultrasonic oscillation action, the CFRTP chopped sheet got more compressed, and the interspace between the prepreg layer was excluded. When the welding energy arrived at 700 J, the PA6 film thickness decreased to 93 μm. It indicates that the PA6 film got softened under 700 J, and with the compression action applied by the horn pressure, the PA6 film thickness decreased. In addition to the softening phenomenon of the PA6 film, the sectional area started to melt under the 700 J welding energy. As the microstructure analyzed in Figure 8 indicates, the 700 J is the starting energy point of the effective joining. With the increase of the welding energy, effective joining can be achieved around interface 1, but due to the squeezing phenomenon, the PA6 film and CFRTP chopped sheet thickness significantly decreased. 

From the above analysis, the effect of the welding energy on the joint at room temperature can be divided into three modes: compaction mode, joining mode, and damage mode. When the welding energy was less than 500 J, it belonged to the compaction mode. In this mode, the energy heat is insufficient to generate effective joining, the PA6 film was kept at the initial status, and CFRTP chopped sheet was compressed only. During the 700 J to 900 J welding energy, the PA6 and the CFRTP chopped sheet can somewhat generate effective joining. In the damage mode, welding energy higher than 1100 J, carbon fiber fractures occurred on the surface following the serious squeezing phenomenon of the PA6 interlayer and carbon fibers, indicating that the UPW energy should be limited to 900 J when applied to bare carbon fibers. 

### 3.2. Effect of the Co-Action of Preheating Temperature and Welding Energy

From the above experiments carried out at room temperature, it can be seen that solely welding energy cannot generate the effective joining of the SPCC-CFRTP hybrid joint with the addition of PA6 intermediate layer; the crevices defects occurring at interface 2 has a severe impact on the joint properties, and it cannot be eliminated by solely increasing the welding energy. In addition, the joining process window obtained under the only action of ultrasonic joining energy was very narrow. Therefore, eliminating crevice defects and enhancing micro-mechanical interlocking is essential to improve joint mechanical properties. 

An external heater instrument was applied with the UPW system to solve these shortcomings to generate the SPCC-CFRTP hybrid joint. The following experiments, listed in Figure 15, were carried out to investigate the effect of the preheating temperature and welding energy and the synergistic result. The previous results showed that welding energy should be limited to 900 J in the following experiments to avoid carbon fiber damage. Under the combined action of ultrasonic welding energy and preheating temperature, the addition of the PA6 film contributed to the generation of the SPCC-CFRTP hybrid joint. 

The co-effect of the preheating temperature and welding energy on the hybrid joint’s mechanical performance was evaluated utilizing lap shear tests. Figure 16 reports the shear strength results under different joining conditions. It depicts that when the preheating temperature was lower than 80 °C, and the welding energy was less than 500 J, the joining components did not generate adhesion and separated from each other, which led to zero lap shear strength of the joint. With the increase of preheating temperature and welding energy, sectional joining began to appear among the joining components, and the lap shear strength started improving. However, the joint’s most giant lap tensile strength is less than 400 N when the preheating temperature is lower than 160 °C. The joint lap shear strength sharply increased when the preheating temperature reached 200 °C. The best joint strength was obtained when the preheating temperature reached 240 °C, and the maximum value was around 1300 N. Meanwhile, the tensile test had almost the same matter under different welding energy, indicating the welding energy did not play an essential role in the joint mechanical performance under 240 °C preheating temperature.

According to the lap shear test results, the conditions of the joints obtained under different welding energy and preheating temperatures can be divided into three categories. The tests in the yellow frame in Figure 17 belong to category 1, the tests in the blue frame belong to category 2, and the red frame is category 3. Category 1 had the worst mechanical performance, the SPCC metal component was completely separate from the other parts, and only a tiny area of PA6 film was fused with the CFRTP chopped sheet. The dashed red line in Figure 18 shows the representative fracture surface morphology, which represents the joining area between the PA6 film and the CFRTP chopped sheet. Only a tiny space was melted between the PA6 film and CFRTP chopped sheet and got adhesive under the welding energy oscillation effect, and under the different joining energy, the joining area was different. It indicates that the lowest energy did not adequately join the hybrid adherents.

In category 2, the lap shear strength started to increase with the improvement of preheating temperature and welding energy. Figure 19 shows the representative fracture surfaces of the hybrid joints of category 2 and finds that the welding zone increased with preheating temperature and welding energy. It indicates that the increased welding zone of direct contact between the PA6 film and the SPCC is responsible for the slightly increased mechanical performance of the SPCC-CFRTP hybrid joints. In this joining category, preheating can change the mechanical properties of the CFRTP plastic matrix, lowering Young’s modulus and softening, so that the ultrasonic deformation and melting of the matrix can easily occur with less welding energy. Approximate joint strength could be obtained under different joining conditions for some cases, such as t12 and t14, showing almost the same strength value. These cases’ tendency shows that preheating temperature reduces the welding energy to a certain extent. The observation of the joint’s microstructure found that the crevices still existed but presented a decreasing tendency with the growth of the total input energy. Therefore, adding the interlayer can enhance the joints’ mechanical strength due to the larger welding area and fewer crevices. 

A significant increase in joint strength was observed when the welding condition belonged to category 3. When the preheating temperature reached 200 °C, the PA6 film started to melt before the joining process but could not be melted totally. Therefore, the viscosity of the molten and softened PA6 film favors the metal surface’s wettability. Under the co-action of welding energy, the adhesion forces are finally applied to the joint. However, due to the insufficient melting of PA6 film under 200 °C preheating temperature, welding energy still played an essential role in the joint strength. As the welding energy increased, the melting mount of the PA6 film increased and became more compressed under the oscillation effect, which in turn led to an increase in the joint strength. This joining condition produces joints not only with higher mechanical strength but smaller amounts of microstructural defects. Microstructural analysis proves that crevices that occurred at interface 2 have been eliminated under this joining condition, as shown in Figure 20. The PA6 film co-cured with the CFRTP chopped sheet has an active role in the bonding mechanism during the ultrasonic joining, and an effective micro-interlocking was achieved as a result of the 200 °C preheating. 

Unlike the sectional melting under 200 °C preheating temperature, the PA6 film can be fully melted at 240 °C before welding. Therefore, the depression area on the SPCC surface can be filled by the melted PA6 film under the action of welding energy oscillation and welding force, and an effective micro-mechanical interlocking could be established at interface 2. Figure 21 shows the joints’ cross-section microstructures with no crevice defects generated. However, due to the different welding energy, the compression amount of the PA6 film was different. With the increase of the welding energy, the more melted PA6 film could be squeezed to the joint edge, leading to decreased distance between carbon fibers and the SPCC. Figure 22 is the representative fracture surface obtained under 240 °C. It enabled to understand that the large welding zone allows good joints mechanical results. 

From the above analysis, it can be found that the preheating temperature dramatically affects the hybrid joint interfacial zone quality. The crevices defects at interface 2 are responsible for the poor joint strength. The microstructural analysis demonstrated that the wide spreading of the molten PA6 film under the high preheating temperature can fill the depression area on the SPCC surface, which generates the micro-mechanical interlocking between the joining components. The joint lap shear strength is thus increased. In addition, the severe squeezing phenomenon of carbon fibers was avoided due to the preheating temperature. 

## 4. Conclusions

Ultrasonic plastic welding joints of CFRTP chopped sheet and SPCC with additional PA6 film intermediate layer were prepared and analyzed regarding their microstructures and mechanical strength. It showed the feasibility of the ultrasonic plastic welding process and optimized the joining process. From the above analysis and discussion, the following conclusions can be drawn. 

(1) Nylon 6 film can be used as an intermediate layer and ultrasonically welded to obtain an effective metal and bare carbon fiber hybrid joint through the optimized joining technology. The melting of the PA6 film is the key to the fusion of carbon fibers and PA6 film, while the widespread spreading of the melted PA6 film on the depression area of the metal surface guarantees the final joint strength. 

(2) With the sole effect of ultrasonic welding energy at room temperature, the bare carbon fibers can have a good fusion with the PA6 film, but it cannot generate an effective joining between metal and the PA6 film. Insufficient joining, narrow welding process windows, and carbon fiber damage at high welding energy are the primary defects that cause poor joint strength.

(3) The joining condition between the metal and the PA6 film is essential to ensure joint strength. The micro-mechanical interlocking generated on the rugged metal surface contributes to the mechanical performances of the ultrasonically welded hybrid joints. 

(4) Effective preheating can reduce crevice joining defects at the SPCC and the PA6 film boundary, guaranteeing the molten PA6 film’s adequate spreading on the depression area on the metal surface and improving the welded joint’s mechanical strength. Furthermore, preheating can reduce the welding energy, which is beneficial for the severe squeezing phenomenon and avoids carbon fiber damage. 

## Figures and Tables

**Figure 1 polymers-14-05235-f001:**
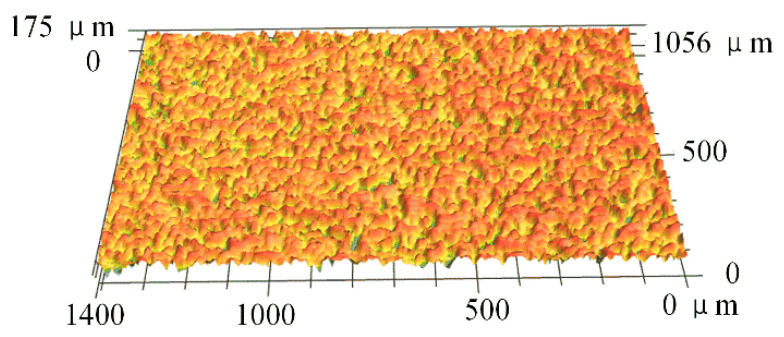
SPCC surface morphology.

**Figure 2 polymers-14-05235-f002:**
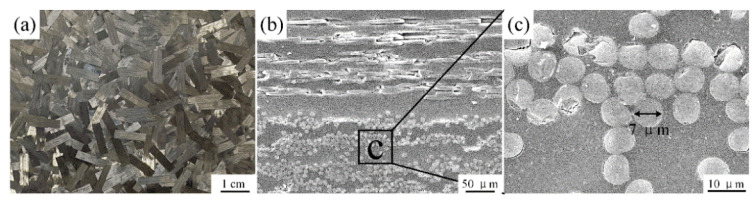
CFRTP chopped sheet: (**a**) macro-structure of the chopped CFRTP sheet from the top view; (**b**) microstructure of the cross-section of the CFRTP chopped sheet; (**c**) carbon fiber diameter.

**Figure 3 polymers-14-05235-f003:**
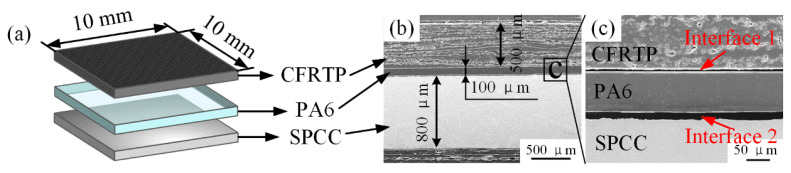
Joint structure composition: (**a**) schematic diagram of the joint composition (not in scale); (**b**) cross-section of the joint before joining; (**c**) interface 1 and 2.

**Figure 4 polymers-14-05235-f004:**
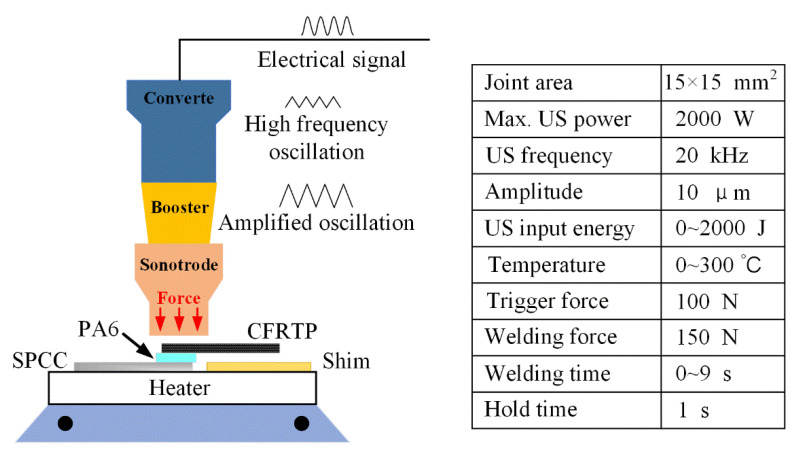
Schematic diagram of the ultrasonic plastic welding system.

**Figure 5 polymers-14-05235-f005:**
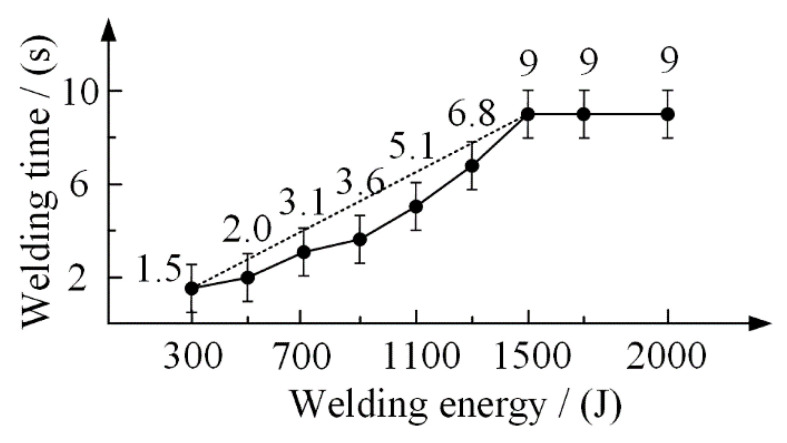
Relationship between welding time and welding energy.

**Figure 6 polymers-14-05235-f006:**
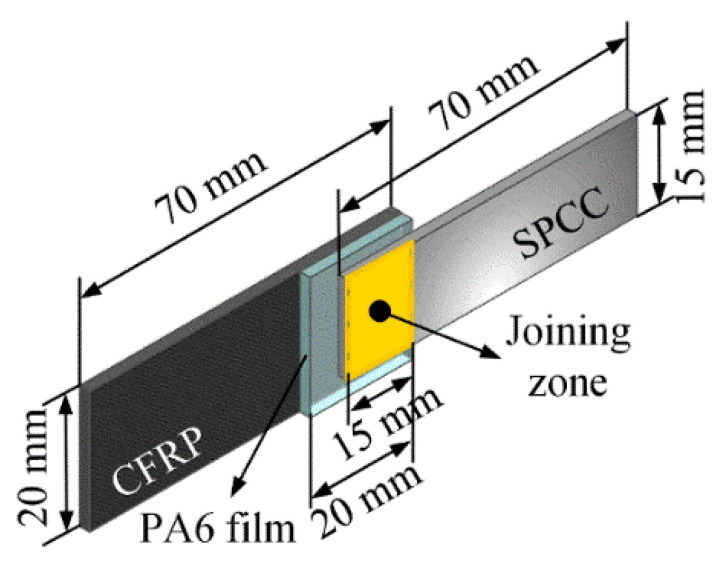
Schematic diagram of the SPCC/CFRP hybrid joint.

**Figure 7 polymers-14-05235-f007:**
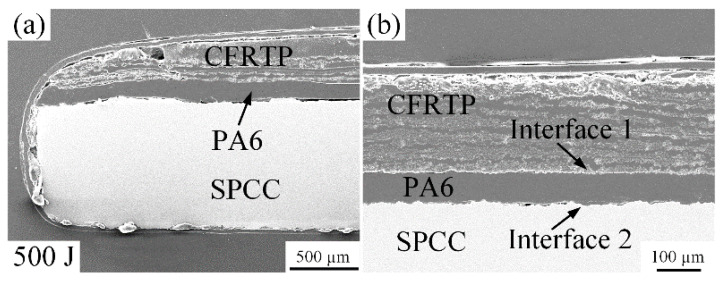
Microstructure of the joint cross-section at the 500 J energy: (**a**) the corner of the joint; (**b**) interface morphology.

**Figure 8 polymers-14-05235-f008:**
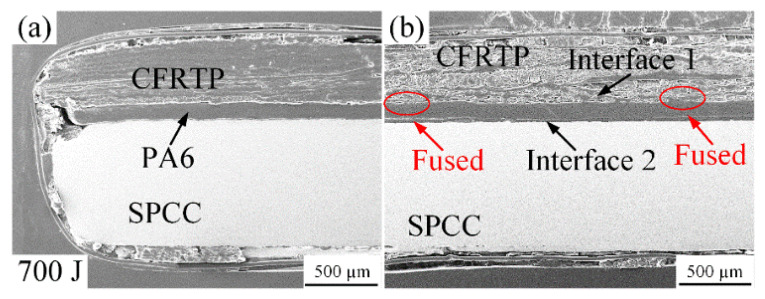
Microstructure of the joint cross-section at the 700 J energy: (**a**) carbon fibers squeezed out phenomenon; (**b**) situation of interface 1 and 2.

**Figure 9 polymers-14-05235-f009:**
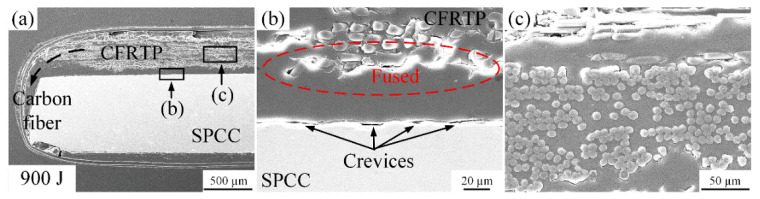
Microstructure of the joint cross-section at the 900 J energy: (**a**) carbon fibers squeezed out phenomenon; (**b**) situation of interface 1 and 2; (**c**) carbon fibers.

**Figure 10 polymers-14-05235-f010:**
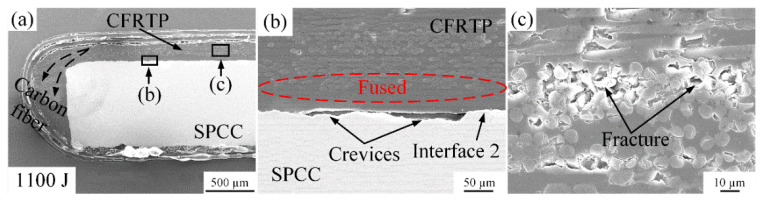
Microstructure of the joint cross-section at the 1100 J energy: (**a**) carbon fibers squeezed out phenomenon; (**b**) situation of interface 1 and 2; (**c**) fractures on carbon fiber.

**Figure 11 polymers-14-05235-f011:**
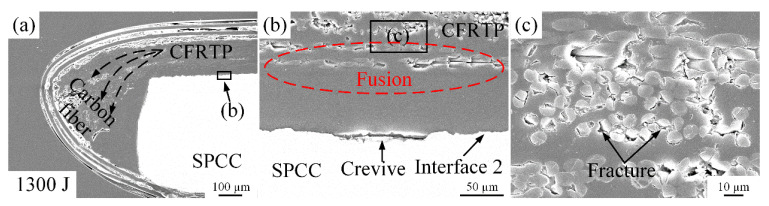
Microstructure of the joint cross-section at the 1300 J energy: (**a**) carbon fibers squeezed out phenomenon; (**b**) situation of interface 1 and 2; (**c**) fractures on carbon fiber.

**Figure 12 polymers-14-05235-f012:**
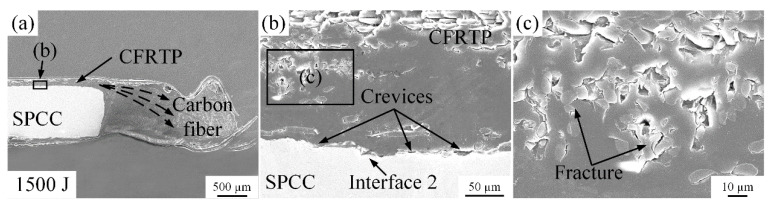
Microstructure of the joint cross-section at the 1500 J energy: (**a**) carbon fibers squeezed out phenomenon; (**b**) situation of interface 1 and 2; (**c**) fractures on carbon fiber.

**Figure 13 polymers-14-05235-f013:**
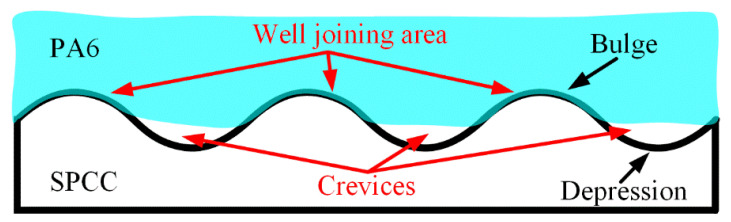
Schematic diagram of the joining result by the sole role of UPW energy.

**Figure 14 polymers-14-05235-f014:**
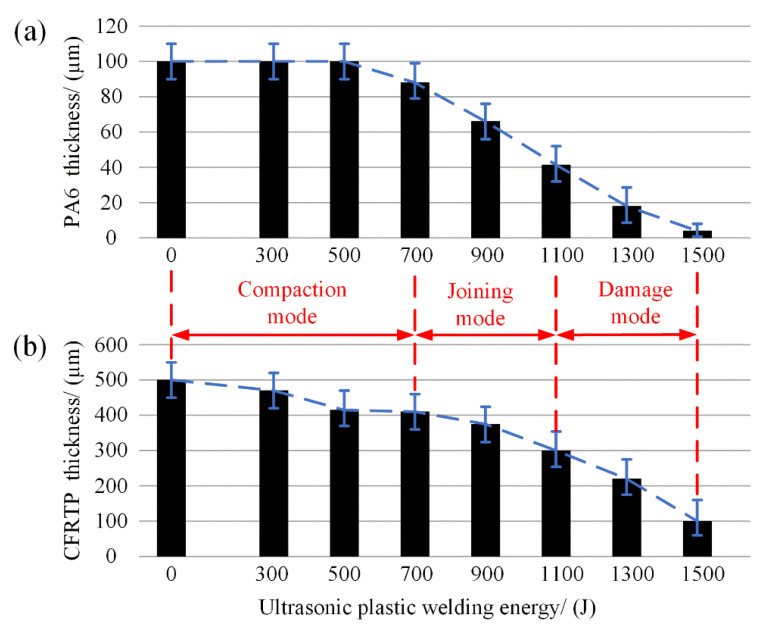
The trend of the thickness variation: (**a**) PA6 film; (**b**) CFRTP chopped sheet.

**Figure 15 polymers-14-05235-f015:**
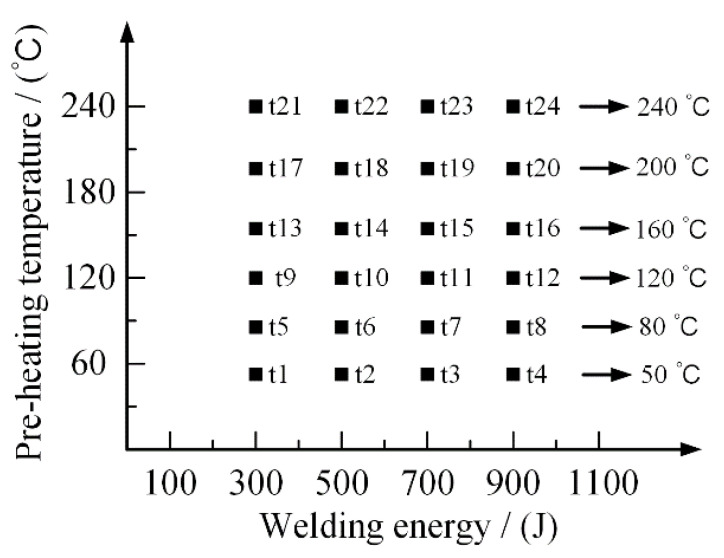
Design of the experiments.

**Figure 16 polymers-14-05235-f016:**
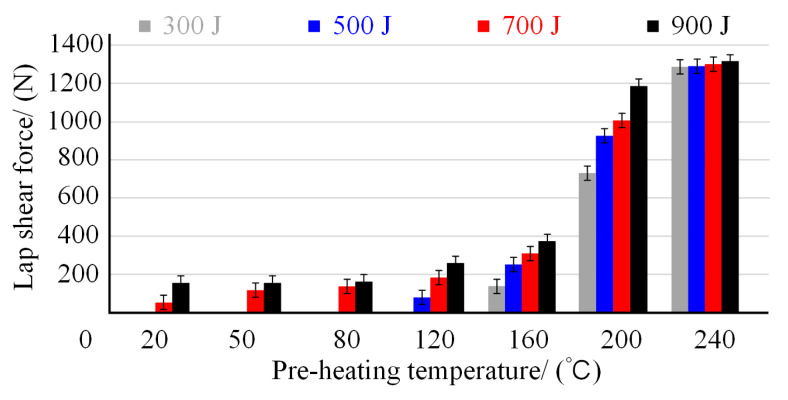
Ultimate lap shear strength of the joints under different joining parameters.

**Figure 17 polymers-14-05235-f017:**
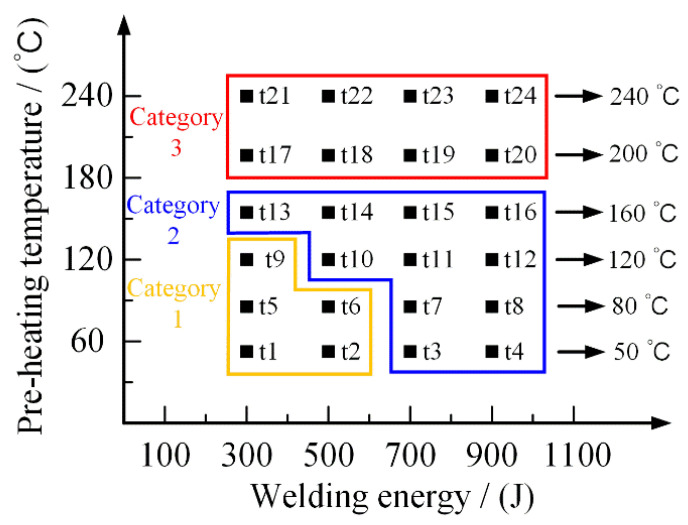
Joining process window of the SPCC-CFRTP hybrid joint by the UPW.

**Figure 18 polymers-14-05235-f018:**
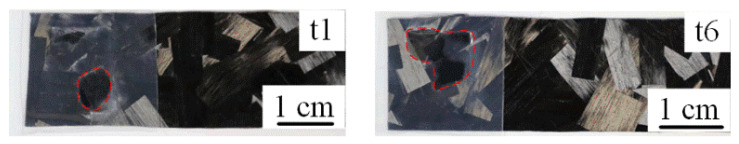
Representative fracture surface of category 1.

**Figure 19 polymers-14-05235-f019:**
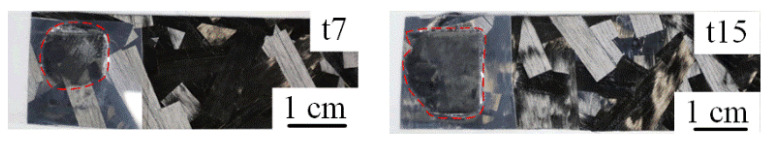
Representative fracture surface of the category 2.

**Figure 20 polymers-14-05235-f020:**
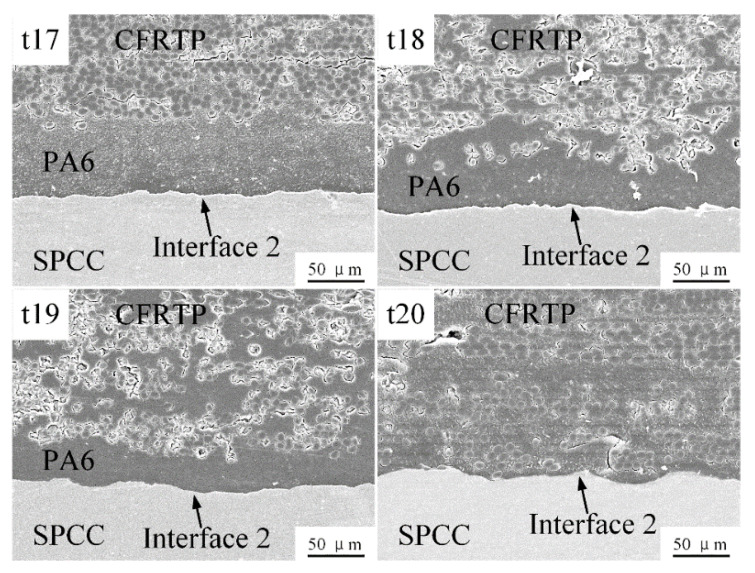
Microstructures of interface 2 obtained under 200 °C.

**Figure 21 polymers-14-05235-f021:**
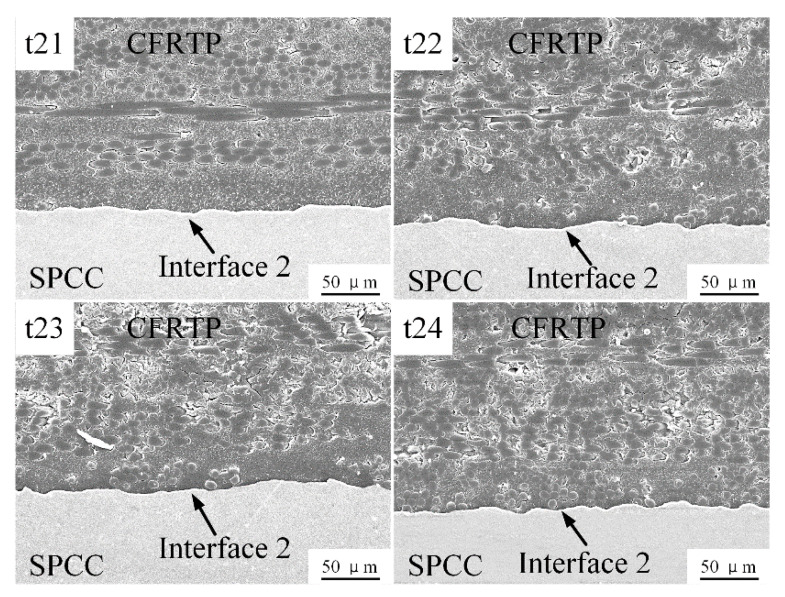
Microstructures of interface 2 obtained under 240 °C.

**Figure 22 polymers-14-05235-f022:**
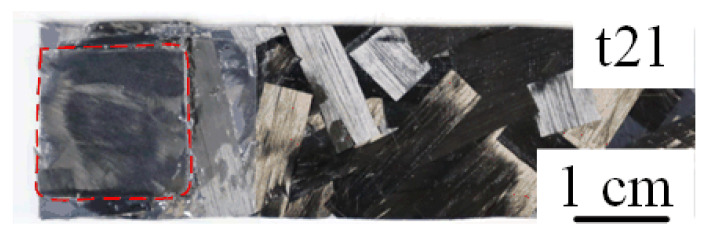
Representative fracture surface of category 3.

**Table 1 polymers-14-05235-t001:** CFRTP chopped sheet properties.

Fiber Volume Content	Resin Content	Thickness	Mass
50 (%)	39 (wt.%)	500 μm	500 g/m^2^

**Table 2 polymers-14-05235-t002:** Nylon 6 properties.

Chemical Formula	Density	Melting Point	Autoignition Temperature
(C_6_H_11_NO)_n_	1.084 g/mL	220 °C	434 °C

## Data Availability

The data presented in this study are available on request from the corresponding author.

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
