# Peer review of "Study on the SPCC and CFRTP Hybrid Joint Performance Produced with Additional Nylon-6 Interlayer by Ultrasonic Plastic Welding"

_polymers, 2022, doi:10.3390/polym14235235_

Round 1

Reviewer 1 Report

The SPCC and CFRTP were joined via UPW with a preheating system, and an intermediate layer was added to improve the joint quality. The effect of the welding energy and preheating temperature on the mechanical properties as well as the macro/microstructure of the joints was analyzed to reveal the joining mechanism and finally obtain a sound joint. The results are interesting and the analysis is reasonable. The manuscript is well organized and contains valuable information. However, the following comments still need to be addressed:

Q1: Please further illustrate the heating principle in detail of the preheating system.

Q2: Please explain why the welding time was a constant 9 s when increasing the welding energy from 1500 J to 2000 J, while the welding time gradually increased with the welding energy ranging from 300 J to 1500 J? (In Fig. 5)

Q3: As the authors mentioned, PA6 was melted under the welding energy higher than 700 J. Please explain what kind of microstructure could be regarded as the symbol of melting in this study?

Q4: What was the predominant difference between SPCC and CFRPT (for example the roughness of the materials) leading to the fact that CFRPT can be successfully joined to PA6 with the sole effect of ultrasonic welding energy at room temperature whereas crevices always existed at the interface between PA and SPCC under the same welding condition?

Q5: Please further explain why “the presence of pre-heating temperature reduces the welding energy to a certain extent in joining category 2” as mentioned in 3.2.

Q6: It is recommended that a figure illustrating the relationship between the strength and the welding energy at room temperature could be added in 3.1 rather than displaying these information in Fig. 16.

Reviewer 2 Report

Dear authors,

thank you very much for a nice manuscript. As a polymer researcher, I control the article mainly from the view of polymer chemistry.

At first, I have some comments for style and very small details.

All figures should have the same format of marking. Somewhere you used Fig. somewhere figure. Please do it same by all figures.

Page 3, line 129: There should be probably the symbol of micrometre so not um.

Page 4, line 131: There should be probably the symbol of micrometre so not um.

Page 5, figure 4: It is amplitude not amplitute.

Page 6, line 200: The testing machine is Zwick/Roell, not Zwich/Roell.

Then I have some comments on the content:

1) You worked with PA6. There was no information about melting temperature, which was very important for your research. Please add there some information about PA6, that can be connected with your research.

2) In methods, there is information about the pre-preg of carbon nanofibers, which are in matrix resin. Which resin?

3) Can the resin play some role in your research? I think sure.

4) There is no information about the curing of the resin, how the curing goes, etc.

5) I think, there could be also a measurement of a combination of SPCC and CFRTP without PA6, to see what happens.

Thank you one more time for a nice manuscript.

Reviewer 3 Report

An interesting research was conducted on SPCC and CFRTP hybrid joint performance with additional Nylon-6 interlayer by ultrasonic plastic welding. This paper is well designed and some significant data are obtained. To improve the quality of the paper, the following comments should be further answered. 

1. Abstract, some quantitative results and analysis should be further supplemented on the interface joining performance and parameter effect.

2. This paper focus on the interface joining performance of metal-carbon fiber reinforced thermoplastics (CFRTP) hybrid structure. However, some advantages and properties of carbon fiber reinforced thermoplastic composites have not been fully summarized. Why do you choose carbon fiber and thermoplastic resin? The authors should analyze the advantages of carbon fiber compared to other fiber types, and the advantages of thermoplastic resin compared with thermosetting resin. Furthermore, the significance of selecting carbon fiber reinforced thermoplastic resin in this paper should be highlighted, so that readers can better understand this material. Please review some research on the two types of composites: Polymer Testing 90, 106761. Polymers, 2019, 11(3): 407. Polymers 14 (6), 1087.

3. The authors selected PA6 film as the thermoplastic intermediate layer in this research. Although PA6 has some good performance, its water uptake rate is generally high. When exposed to hygrothernal environment, swelling effect from water uptake may cause the interface debonding. How do the authors consider this problem?

4. Part 2.1.1-2.1.3, please provide some basic mechanical performance parameters of raw materials.

5. In Fig. 2b, it can be seen that there are some typical cracks in the upper part of the picture. Are these cracks caused by grinding or other reasons?

6. Part 2.2.1, during the jointing, is the SPCC surface specially treated? What is the treatment process?

7. In Figure 4, it can be seen that the thickness of PA6 film and CFRTP material are decreasing with the increase of welding energy. Is there an optimal welding energy and thickness? How to determine these two indicators?

8. Figure 15 and 17 (the experimental design) should be included in Part 2.

9. It can be found that when the preheating temperature is 240oC, the ultimate lap shear strength of the joints is the highest. Is this temperature related to the melting point temperature of PA6 resin?

Round 2

Reviewer 3 Report

It can be accepted.